# Impact of community piped water coverage on re-infection with urogenital schistosomiasis in rural South Africa

**Polycarp Mogeni**[1,2,3]*, **Alain Vandormael**[1,2,3,4], **Diego Cuadros**[5,6], **Christopher Appleton**[7], **Frank Tanser**[1,2,8]

[1]Africa Health Research Institute, KwaZulu-Natal, South Africa; [2]School of Nursing and Public Health, University of KwaZulu-Natal, KwaZulu-Natal, South Africa; [3]KwaZulu-Natal Innovation and Sequencing Platform (KRISP), University of KwaZulu-Natal, KwaZulu-Natal, South Africa; [4]Heidelberg Institute of Global Health, Faculty of Medicine, University of Heidelberg, Heidelberg, Germany; [5]Department of Geography and Geographic Information Science, University of Cincinnati, Cincinnati, United States; [6]Health Geography and Disease Modeling Laboratory, University of Cincinnati, Cincinnati, United States; [7]School of Life Sciences, University of KwaZulu-Natal, KwaZulu-Natal, South Africa; [8]Lincoln International Institute for Rural Health, University of Lincoln, Lincoln, United Kingdom

**Abstract** Previously, we demonstrated that coverage of piped water in the seven years preceding a parasitological survey was strongly predictive of *Schistosomiasis haematobium* infection in a nested cohort of 1976 primary school children (Tanser, 2018). Here, we report on the prospective follow up of infected members of this nested cohort (N = 333) for two successive rounds following treatment. Using a negative binomial regression fitted to egg count data, we found that every percentage point increase in piped water coverage was associated with 4.4% decline in intensity of re-infection (incidence rate ratio = 0.96, 95% CI: 0.93–0.98, p=0.004) among the treated children. We therefore provide further compelling evidence in support of the scaleup of piped water as an effective control strategy against *Schistosoma haematobium* transmission.

*For correspondence:
pkambona11@gmail.com

**Competing interests:** The authors declare that no competing interests exist.

## Introduction

About 243 million people are infected with schistosomiasis worldwide, of whom ~ 93% reside in sub-Saharan Africa where children carry the greatest burden of the disease. In the tropics and subtropics, schistosomiasis is a major cause of disability among neglected tropical diseases (NTDs) accounting for 1.43 million disability-adjusted life-years lost in 2017 (*GBD 2017 Disease and Injury Incidence and Prevalence Collaborators, 2018*). Infection occurs when trematodes of the genus Schistosoma shed by infected freshwater snails (an intermediate host) penetrate the skin upon contact with infested water (*Colley et al., 2014*). Intensity of infection in the human host is a function of the parasite load and can indirectly be quantified by the number of eggs excreted. Host variations in worm burden has been attributed to recent chemoprophylaxis, heterogeneities in exposure and host susceptibility (*Colley et al., 2014*). Most human infections in sub-Saharan Africa (SSA) are due to *Schistosoma mansoni*, which causes intestinal schistosomiasis and *Schistosoma haematobium* responsible for urogenital schistosomiasis (*Lai et al., 2015*). However, *Schistosoma haematobium* has the widest geographical coverage in SSA and is the main cause of infection in the Hlabisa sub-district, where our study is based (*Saathoff et al., 2004*).

In our previous work, we assessed the impact of piped water coverage on the risk of *Schistosoma haematobium* infection in a rural South African community (*Tanser et al., 2018*). We argued that a measure of piped water access by individuals/households, which is commonly used in the existing literature (*Grimes et al., 2014*), is less sensitive than a measure of piped water coverage at the community level. Our hypothesis is that a higher community coverage of piped water reduces both an individual's exposure to parasitic agents (direct benefit) as well as the number of contacts that infectious individuals in the surrounding have with open water bodies, thus decreasing the overall transmission intensity of *Schistosoma haematobium* within the community (indirect benefit).

We previously used novel geostatistical methods, annual population-based surveillance data, and a parasitological survey to quantify the risk of *Schistosoma haematobium* infection in a nested cohort of 1976 primary school children (*Tanser et al., 2018*). In this baseline parasitological survey, we showed that every percentage increase in community piped water (in the prior 7 years) was associated with a 2.5% decrease in the odds of *Schistosoma haematobium* infection. However, we did not determine if community piped water coverage reduced re-infection rates among the same children who were treated with praziquantel during the baseline survey.

Here, we report the results of two consecutive rounds of follow-up of infected children, which were undertaken between September and December of 2007 (round 1), and between April and August of 2008 (round 2) respectively. To the best of our knowledge, this is the first study to systematically evaluate the impact of community piped water coverage on *Schistosoma haematobium* re-infection rates following treatment with praziquantel and therefore address a major public health evidence gap highlighted in a recent review (*Grimes et al., 2015*). A strong relationship would provide compelling evidence for the protective effect of increased piped water coverage and have broad implications for the treatment and management of *Schistosoma haematobium* in resource-limited settings.

## Results

During the baseline parasitological survey, a total of 2105 children from all 33 primary schools located in a contiguous geographical area in rural KwaZulu-Natal consented to participate in the study. Of these participants, 1976 were residents in the study area (*Figure 1*). The prevalence of baseline infection was 16.9% (95%CI: 15.2–18.6) (*Figure 1*). Further detailed baseline characteristics of the study participants and baseline analyses are presented in our previously reported findings (*Tanser et al., 2018*).

### Re-infection cohort characteristics

Out of the 333 microscopically confirmed infections at baseline, 253 (76%) consented to screening for *Schistosoma haematobium* re-infection at round 1 and 125 consented for screening at round 2 (*Figure 1*). The prevalence of light re-infection and heavy re-infection was 15.4% (95%CI: 11.2–20.5) and 8.7% (95%CI: 5.5–12.9) for round 1, and 12.8% (95%CI: 7.5–20.0) and 6.4% (95%CI: 2.8–12.2) for round 2, respectively. The geometric mean egg counts were 16.7 (95%CI: 9.4–29.6) eggs/10 mL for round 1 and 18.2 (95%CI: 6.5–50.6) eggs/10 mL for round 2. Detailed characteristics of study participants for each follow-up round are presented in *Table 1* and *Figure 2A*.

In the pooled analysis (n = 378), 119 (31.5%) were girls and 69 (18.3%) were below 11 years of age. Overall the rate of re-infection was 36 (95%CI: 27–48) infections/100 person-years of follow-up. The median community piped water coverage in 2007 was 91.2% (inter quartile range (IQR), 71.0–97.7) (*Figure 2B*) and was geographical heterogeneous (*Tanser et al., 2018*). As we previously noted, the proportion of children with heavy infection at baseline increased from 5.1% (95%CI: 1.9–10.7) among children $\leq$ 9 years old to 14.8% (95%CI: 10.3–20.4) among children $\geq$ 14 years of age ($\chi^2$-test-of-trend=22.96, p<0.001) (*Tanser et al., 2018*). In contrast, in the re-infection cohort, the proportion of heavy re-infection decreased with increasing age, from 13.0% (95%CI: 6.1–23.3) among children < 11 years of age to 6.9% (95%CI: 2.8–13.8) among children > 12 years of age ($\chi^2$-test-of-trend=3.317, p=0.3454) although with lower statistical power. The decline in re-infection prevalence towards older children was more marked in girls than boys (*Figure 3A and B*).

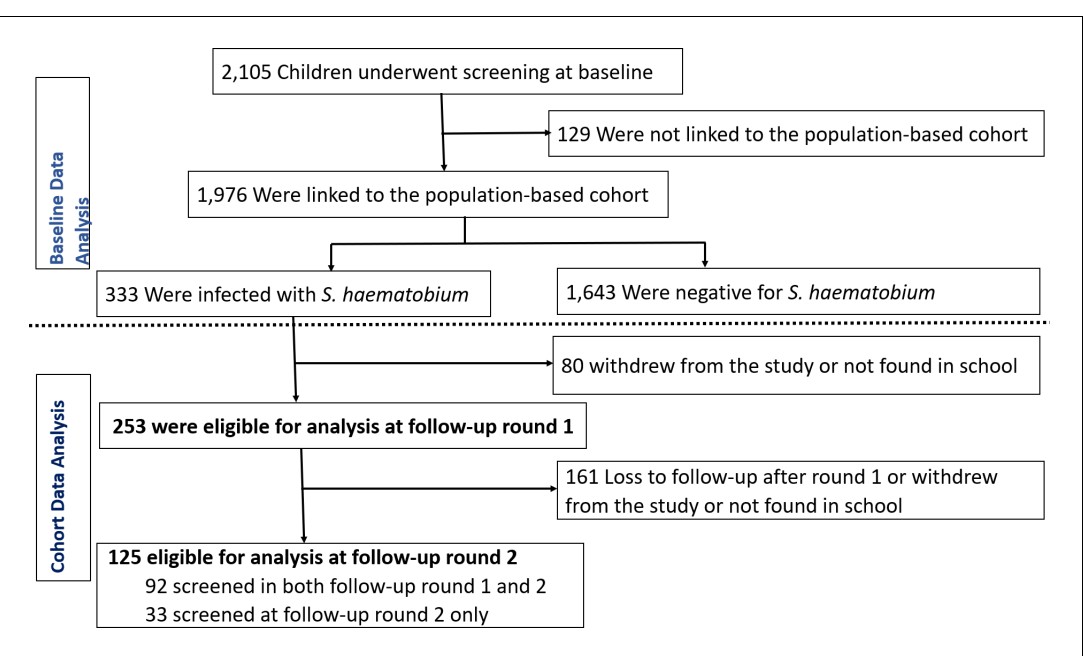

**Figure 1.** Baseline screening for *Schistosoma haematobium* infection and re-infection follow-up rounds of the study participants. Participants who were not linked to the population-based cohort study were excluded from the baseline analysis presented in our previous analyses (*Tanser et al., 2018*) and those not treated for infection at baseline were excluded from the re-infection analysis. Participants who were treated at baseline but not screened at round 1 were eligible for screening at round 2 if they provided informed consent.

## Impact of community piped water coverage on intensity of re-infection

In the pooled adjusted negative binomial model (n = 378), a percentage point increase in community piped water coverage in the area surrounding a child's residence was associated with 4.4% decline in mean re-infection intensity (Model 2 in *Table 2*; incidence rate ratio (IRR) = 0.96, 95% CI: 0.93–0.98, p=0.004). Lower piped water coverage areas were associated with significantly higher mean egg counts albeit with relatively low numbers of children living at low piped water coverages (*Figure 4*).

An analysis of each follow-up survey round separately (N = 253 and N = 125 for round 1 and round 2 respectively) revealed a consistent pattern in which community piped water was strongly protective against *Schistosoma haematobium* re-infection intensity. That is, every percentage increase in community piped water coverage was associated with ~3% and~8% decrease in intensity of re-infection at follow-up round 1 and 2 respectively (Model 2 of *Supplementary file 1* and Model 2 of *Supplementary file 2*). In both follow-up rounds, boys were at a higher risk of intense re-infection than girls (Model 2 of *Supplementary file 1*, and Model 2 of *Supplementary file 2*).

Details of cohort retention are provided in *Supplementary file 3* and *Supplementary file 4*. Dropout was higher among participants residing away from water bodies (>2 km) in follow-up round 1 and among girls in follow-up round 2. However, there was no clear pattern between dropout and piped water coverage.

## Clusters of high re-infection intensity

In a pooled analysis (n = 378), the weighted Gaussian kernel density estimation revealed marked geographical heterogeneity in the geometric mean egg counts across the study area (*Figure 5*). In addition, we detected one significant cluster of higher than average geometric mean egg count (radius = 6.93 km, geometric mean egg count = 54.95, p=0.006) near the Mfolozi river in the southeastern part of the study area. This cluster partially overlapped with one of the existing clusters detected in our baseline analysis (*Tanser et al., 2018*).

**Table 1.** Characteristics of children enrolled in the re-infection cohort.

| | Follow-up round 1 (N = 253) | | | Follow-up round 2 (N = 125) | | |
|---|---|---|---|---|---|---|
| | Total | Infected n(%) | (95% CI) | Total | Infected n(%) | (95% CI) |
| Overall | 253 | 61 (24.1) | (19.0–29.9) | 125 | 24 (19.2) | (12.7–27.2) |
| Gender | | | | | | |
| Female | 86 | 16 (18.6) | (11.0–28.4) | 33 | 5 (15.2) | (5.1–31.9) |
| Male | 167 | 45 (27.0) | (20.4–34.3) | 92 | 19 (20.7 | (12.9–30.4) |
| Age group | | | | | | |
| ≤10 | 41 | 13 (31.7) | (18.1–48.1) | 28 | 7 (25.0) | (10.7–44.9) |
| 11 | 71 | 15 (21.1) | (12.3–32.4) | 34 | 5 (14.7) | (5.0–31.1) |
| 12 | 74 | 18 (24.3) | (15.1–35.7) | 29 | 6 (20.7) | (8.0–39.7) |
| ≥13 | 67 | 15 (22.4) | (13.1–34.2) | 34 | 6 (17.6) | (6.8–34.5) |
| Community piped water coverage (%) | | | | | | |
| <70 | 58 | 17 (29.3) | (18.1–42.7) | 31 | 5 (16.1) | (5.5–33.7) |
| 70 - < 90 | 66 | 13 (19.7) | (10.9–31.3) | 28 | 8 (28.6) | (13.2–48.7) |
| ≥90 | 129 | 31 (24.0) | (16.9–32.3) | 66 | 11 (16.6) | (8.6–27.9) |
| Altitude class (meters) | | | | | | |
| <50 | 17 | 5 (29.4) | (10.3–56.0) | 14 | 6 (42.9) | (17.7–71.1) |
| 50–100 | 140 | 31 (22.1) | (15.6–29.9) | 62 | 11 (17.7) | (9.2–29.5) |
| 100–150 | 84 | 21 (25.0) | (16.2–35.6) | 42 | 5 (11.9) | (4.0–25.6) |
| 150–200 | 7 | 2 (28.6) | (3.7–71.0) | 3 | 0 (0) | (0–70.8) |
| ≥200 | 5 | 2 (40.0) | (5.3–85.3) | 4 | 2 (50.0) | (6.8–93.2) |
| Distance water body class | | | | | | |
| <1 km | 92 | 20 (21.7) | (13.8–31.6) | 46 | 11 (23.9) | (12.6–38.8) |
| 1–2 km | 98 | 25 (25.5) | (17.2–35.3) | 42 | 8 (19.1) | (8.6–34.1) |
| 2–3 km | 46 | 13 (28.3) | (16.0–43.5) | 26 | 5 (19.2) | (6.6–39.4) |
| >3 km | 17 | 3 (17.7) | (3.8–43.4) | 11 | 0 (0) | (0–28.5) |
| School grade | | | | | | |
| Grade 5 | 144 | 37 (25.7) | (18.8–33.6) | 74 | 13 (17.6) | (9.7–28.2) |
| Grade 6 | 109 | 24 (22.0) | (14.6–31.0) | 51 | 11 (21.6) | (11.3–35.3) |
| Toilet | | | | | | |
| No Toilet | 47 | 13 (27.7) | (15.6–42.6) | 23 | 1 (4.3) | (0.1–21.9) |
| Toilet | 206 | 48 (23.3) | (17.7–29.7) | 102 | 23 (22.6) | (14.9–31.9) |
| Land cover classification | | | | | | |
| Closed shrubland | 145 | 35 (24.1) | (17.4–31.9) | 65 | 12 (18.5) | (9.9–30.0) |
| Open shrubland | 59 | 14 (23.7) | (13.6–36.6) | 34 | 6 (17.7) | (6.8–34.5) |
| Sparse shrubland | 41 | 11 (26.8) | (14.2–42.9) | 19 | 6 (31.6) | (12.6–56.6) |
| Thickett | 8 | 1 (12.50) | (0.3–52.7) | 7 | 0 (0) | (0–41.0) |
| Baseline intensity of infection | | | | | | |
| Light infection | 105 | 35 (33.3) | (24.4–43.2) | 50 | 12 (24.0) | (13.1–38.2) |
| Heavy infection | 148 | 26 (17.6) | (11.8–24.7) | 75 | 12 (16.0) | (8.6–26.3) |
| Sample size (N) | 253 | | | 125 | | |

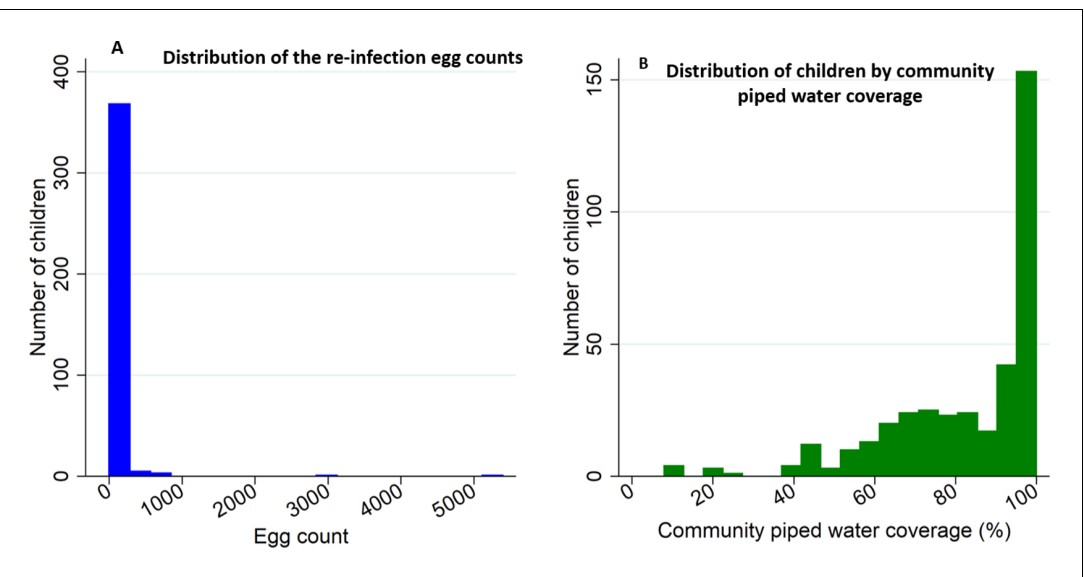

**Figure 2.** Histograms of *Schistosoma haematobium* egg counts and community piped water coverage for the re-infection cohort participants (n = 378). Panel (**A**) shows the distribution of egg counts/10 mL among children observed at follow-up round 1 and 2, and panel (**B**) shows the distribution of community piped water coverage among all study participants. Community piped water coverage in the community surrounding each child was derived from the population-based 2007 piped water use survey conducted in all households in the study area.

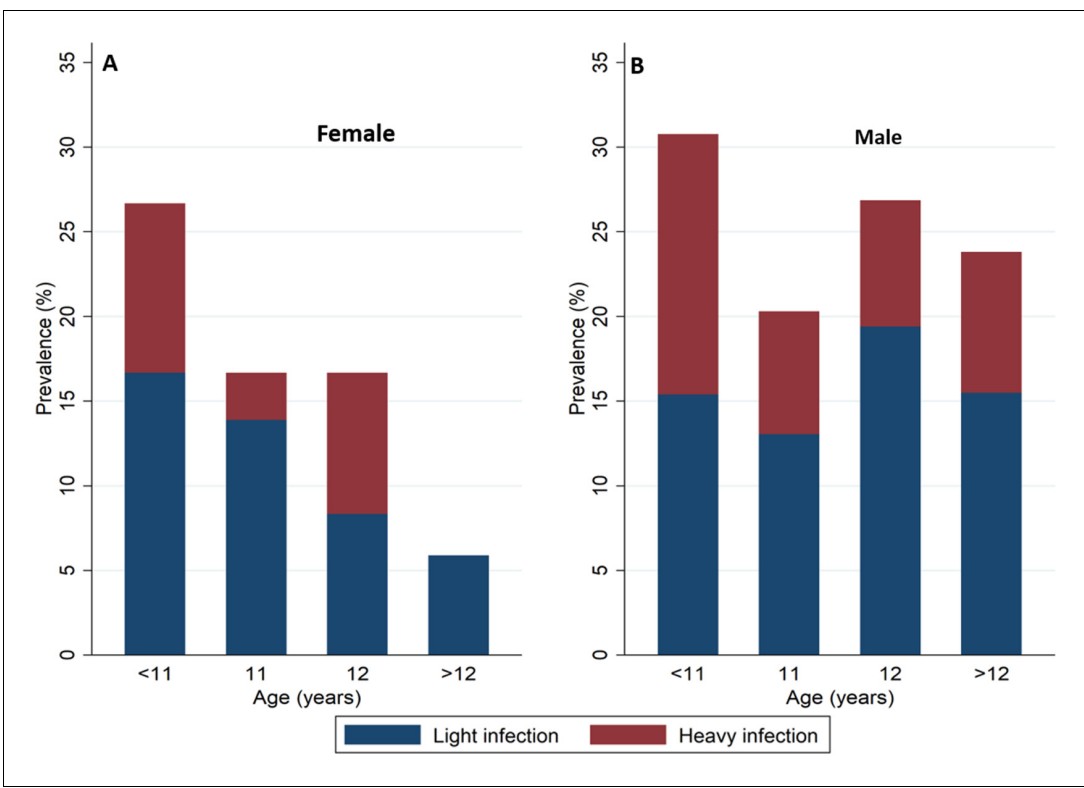

**Figure 3.** Prevalence of *Schistosoma haematobium* re-infection and intensity of re-infection by age and sex among children taking part in the re-infection cohort (n = 378). Blue represents light re-infections (<50 eggs per 10 ml urine) and Red represents heavy re-infections (≥50 eggs per 10 ml urine). Panels **A** and **B** show the prevalence of *Schistosoma haematobium* for female and male children respectively.

# Discussion

Using data from one of Africa's largest population-based cohorts, we previously demonstrated that high coverage of piped water in the seven years preceding a parasitological survey was strongly predictive of *Schistosoma haematobium* infection in a nested cohort of primary school children (*Tanser et al., 2018*). Here we report on the prospective follow up of infected members of this nested cohort for two successive rounds of testing and treatment. Our results demonstrate a large impact of community piped water coverage on intensity of re-infection. Every percentage increase in piped water coverage was associated with 4.4% decrease in re-infection intensity. Taking these findings together with our previously reported baseline analyses (*Tanser et al., 2018*), we conclude that the scaleup of piped water coverage in the local community surrounding a child's residence is strongly protective against *Schistosoma haematobium* infection and re-infection after treatment.

Unsurprisingly, we previously (*Tanser et al., 2018*) noted a strong positive relationship between age and *Schistosoma haematobium* infection prevalence likely due to the increasing cumulative exposure to infested water (and hence infection with *Schistosoma haematobium*) with increasing age (*Dawaki et al., 2016*; *Wami et al., 2014*). In this study and consistent with previous work (*Mbanefo et al., 2014*; *Roberts et al., 1993*), we observed a higher risk of re-infection among the younger age groups where intensity of exposure to contaminated water is likely higher (*Mbanefo et al., 2014*; *Chandiwana et al., 1991*). Evidence has shown that naturally acquired immunity against *Schistosoma haematobium* infection reduces both intensity of infection and infection prevalence in older age groups in endemic areas (*Mitchell et al., 2011*). Whilst developing this protective immunity take time, treatment with praziquantel has been shown to enhance host protective immunity by exposing large quantities of the parasite antigens required to develop immunity (*Roberts et al., 1993*; *Chisango et al., 2019*; *Fukushige et al., 2019*). In this study, naturally acquired immunity may also have played a role in the observed decline in prevalence of re-infection with increasing age following treatment. However, the impact is likely to be minimal given the narrow age range that was examined.

Differences in gender roles resulting from cultural differences may differentially predispose girls (*Gyuse et al., 2010*) or boys (*Chandiwana et al., 1991*; *Liao et al., 2011*; *Geleta et al., 2015*; *Senghor et al., 2015*) to increased contact with infested water therefore increasing their risk of re-infection with *Schistosoma haematobium* after treatment (*Mbanefo et al., 2014*). These differences explain the heterogeneous findings on gender observed across different geographical settings (*Mbanefo et al., 2014*; *Liao et al., 2011*). In our study and consistent with studies conducted elsewhere (*Chandiwana et al., 1991*; *Liao et al., 2011*; *Geleta et al., 2015*; *Senghor et al., 2015*), girls were at a lower risk of intense re-infection compared to boys (*Table 2*).

We detected a significant local cluster of intense re-infection in the current analysis that partially overlaps with the cluster of *Schistosoma haematobium* infection observed in the baseline survey (*Tanser et al., 2018*), demonstrating that exposure to *Schistosoma haematobium* infested water in the study area is heterogeneous and that transmission is concentrated in certain key locations in keeping with evidence from this and other settings (*Brooker, 2007*; *Simoonga et al., 2008*; *Manyangadze et al., 2016*). *Schistosoma haematobium* clusters can potentially be targeted with available control interventions to interrupt transmission (*Simoonga et al., 2008*) and subsequently achieve elimination (*Bergquist et al., 2017*; *Ross et al., 2017*).

Our study had important strengths: firstly, we utilized a cohort of children who were treated for *Schistosoma haematobium* infection at baseline and had two consecutive rounds of followed-up to assess re-infection intensity. This design presents a strong basis to directly quantify the causal association between community piped water coverage and *Schistosoma haematobium* infection. Secondly, the study was nested within a large population-based cohort in rural KwaZulu-Natal province with detailed homestead level geospatial data linking each child to their residence within the demographic surveillance area. Thirdly, we had access to a comprehensive survey of household level piped water use and asset ownership conducted in 2007 that we utilized to derive an index of community piped water coverage and household wealth index. Finally, we utilized longitudinal databases of environmental predictors of disease infection, described in detail in our previous work (*Tanser et al., 2018*), to adjust for potential confounding in the regression models.

A limitation to our study is that we did not assess praziquantel treatment effect after drug administration. It may therefore be difficult to ascertain whether a positive diagnosis was due to re-

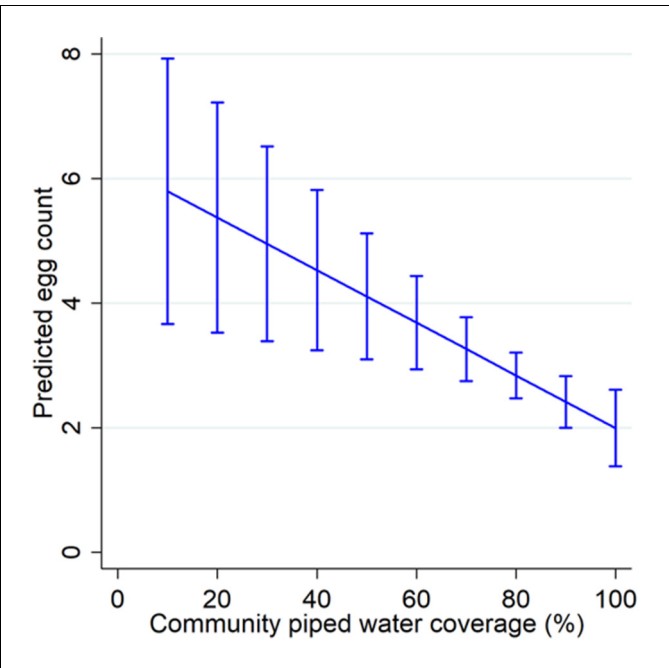

**Figure 4.** Margin plot of piped water coverage and re-infection intensity. The margin plot was constructed from the final parsimonious multivariable negative binomial regression model for the pooled dataset (n = 378, incidence rate ratio = 0.96, p=0.004). Piped water coverage was estimated using the Gaussian kernel density methodology.

infection or treatment failure. However, approximately 80% cure rates at 4 weeks after treatment with praziquantel has been reported in KwaZulu-Natal (*Saathoff et al., 2004*) and Côte d'Ivoire (*N'Goran et al., 2001*), suggesting that most positive diagnoses after treatment were likely re-infections. Furthermore, we observed a shift in burden of heavy infections from older age groups at baseline survey (pre-treatment) to younger age groups in the re-infection cohort analysis that cannot be accounted for by treatment failure, thus providing further evidence suggesting that the impact of treatment failure was minimal. Whilst we demonstrated a clear relationship between piped water coverage and re-infection intensity, the study was not powered to detect differences in the absolute rate of re-infection by piped water coverage category. Furthermore, we observed an increase in the proportion of dropouts with increasing distance from the water bodies (follow-up round 1, *Supplementary file 3*) and among girls (follow-up round 2, *Supplementary file 4*). An increase in dropout rates among individuals who are at a lower risk of infection (that is, residing away from water bodies or girls [*Tanser et al., 2018*]) may partially explain the high re-infection rates documented in follow-up round 1 and follow-up round 2. In our baseline analysis we showed that sex was a strong independent predictor of infection with *Schistosoma haematobium* and that the impact of community piped water coverage was greater among girls than among boys (*Tanser et al., 2018*). Therefore, a higher proportion of attrition among girls will potentially bias the association of piped water on re-infection intensity towards the null hypothesis, implying that our effect estimates may be marginally conservative.

## Conclusion

The WHO recommends mass drug administration (MDA) with a single oral dose of 40 mg/kg of praziquantel for the global control and elimination of schistosomiasis. Although this strategy has clear short-term benefits of reduced morbidity, sustained benefits are uncertain given the current global MDA coverage of circa 20%, low drug compliance and efficacy, and rapid re-infection rates (*Ross et al., 2017*). Our study provides evidence that improved access to piped water in the community significantly reduces the intensity of re-infection among school going children. Therefore, *Schistosoma haematobium* control programs should consider scaleup of piped water

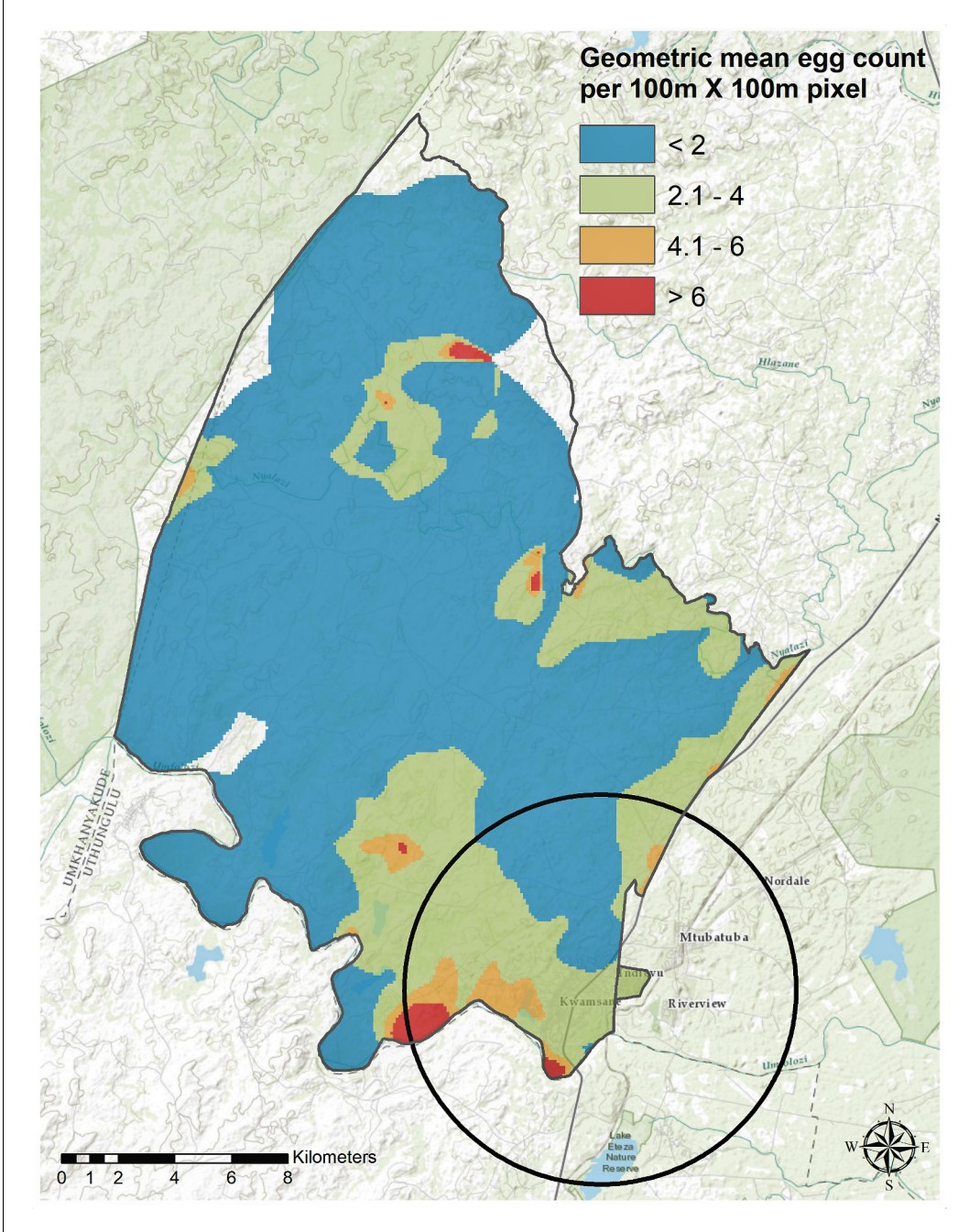

**Figure 5.** Geospatial heterogeneity in *Schistosoma haematobium* geometric mean egg counts (intensity of re-infection) across the study area. The map shows the geographical distribution of mean egg counts/10 mL estimated using the Gaussian kernel of 3 km radius for the pooled re-infection cohort datasets (n = 378). Superimposed on the map is the local cluster (radius = 6.93 km, geometric mean egg count = 54.95, p=0.006) detected using Kulldorff's spatial scan statistic.

coverage in the community as a preventive strategy against re-infection to supplement the MDA strategy. Secondly, targeting households within the high risk areas would mean the most vulnerable population is prioritized with the added benefits of reducing transmission to the entire community (*Lai et al., 2015*; *Bergquist et al., 2017*; *Ross et al., 2017*). We recommend targeting children living along the river Mfolozi and river Nyalazi (cluster locations) to achieve greater impact on reducing morbidity and transmission potential. Finally, we recommend further studies to examine the impact

of an integrated approach (*Knopp et al., 2019*) that include MDA, piped water coverage and behavioral change education (potentially including installation of safe water recreational areas [*Kosinski et al., 2012*]) on re-infection.

## Materials and methods

Ethical approval was provided by the Biomedical Research Ethics Committee of the university of KwaZulu-Natal (reference #E165/05). Written informed consent was sought from parents or guardians of the participating children for both rounds of follow-up in 2007 and 2008 and assent obtained from the children during the follow-up surveys.

### Study site

We undertook our study in all 33 primary schools located within the catchment area of the Africa Health Research Institute (AHRI, previously called the Africa Centre Demographic Information System)(*Tanser et al., 2008*). The study area is located in the coastal lowland area of the northern KwaZulu-Natal province, South Africa (*Figure 6A and B*) and is one of the largest and comprehensive HIV population-based cohorts in sub-Saharan Africa. The surveillance area covers approximately 438 km$^2$ with a population of ~90,000 people living in ~10,000 households. Tri-annual household surveys are conducted by trained field workers under the management of AHRI. Field workers interview a key household informant, who provides information on the births, deaths, migration events, and relationship characteristics of all household members. All households have been geolocated to an accuracy <2 m.

### Re-infection cohort design

The re-infection followup rounds were undertaken between September and December of 2007 (round 1), and between April and August of 2008 (round 2) respectively. The cohort included children who had microscopically confirmed *Schistosoma haematobium* infection at baseline, received treatment for baseline infection and provided informed consent to participate in any of the follow-up rounds. Participants with confirmed infection at baseline or during the follow-up rounds were treated immediately at school. *Schistosoma haematobium* infection status was determined using the urine reagent strips for evaluating microhematuria with confirmation using microscopy diagnosis. A single oral dose of 40 mg/kg body weight of praziquantel was administered to children with detectable microhematuria following recommended World Health Organization (WHO) guidelines

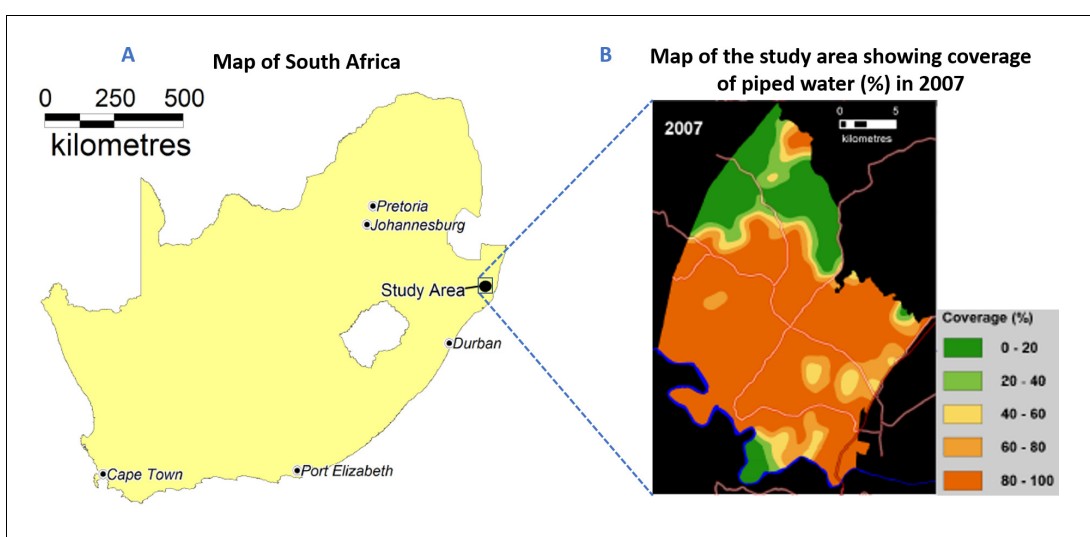

**Figure 6.** Location of the study area in South Africa. Panel A displays the map of South Africa highlighting the major towns and the location of the study area. Panel B displays the map of the study area showing the major roads and the coverage of piped water (%) in 2007. Community piped water coverage was estimated using the Gaussian kernel methodology (*Tanser et al., 2018*).

**Table 2.** Predictors of intensity of re-infection with *Schistosoma haematobium* (pooled analysis, n = 378).
Model 1 presents results from the univariable negative binomial model and Model 2 presents results from the final parsimonious multivariable negative binomial model. Homestead level piped water coverage was derived from a Gaussian kernel density estimation using data from a survey conducted in 2007.

| Covariates | Model 1: univariable (n = 378) | | | Model 2: multivariable (n = 378) | | |
|---|---|---|---|---|---|---|
| | IRR | 95% | P-value | IRR | 95% | P-value |
| Female | 0.17 | 0.06–0.54 | 0.003 | 0.14 | 0.06–0.32 | <0.001 |
| Community piped water coverage (continuous effect) | 0.96 | 0.93–0.98 | 0.002 | 0.96 | 0.93–0.98 | 0.004 |
| Age at baseline (years) | 0.68 | 0.50–0.93 | 0.017 | 0.78 | 0.59–1.04 | 0.094 |
| Altitude class (ref < 50) | | | | | | |
| 50–100 | 3.65 | 0.91–14.5 | 0.067 | 1.20 | 0.31–4.56 | 0.793 |
| 100–150 | 0.72 | 0.21–2.54 | 0.612 | 0.41 | 0.1–1.74 | 0.226 |
| ≥150 | 0.11 | 0.02–0.62 | 0.012 | 0.05 | 0.01–0.32 | 0.001 |
| Land cover class (ref. Sparse shrubland) | | | | | | |
| Closed shrubland | 1.96 | 0.51–7.57 | 0.327 | 0.86 | 0.34–2.21 | 0.754 |
| Open shrubland/grassland | 1.77 | 0.33–9.49 | 0.508 | 1.41 | 0.48–4.16 | 0.533 |
| Thickett | 0.01 | 0.00–0.06 | <0.001 | 0.02 | 0.00–0.20 | 0.001 |
| Toilet in household (ref. no toilet) | 2.71 | 0.70–10.4 | 0.148 | 0.77 | 0.24–2.46 | 0.662 |
| Grade (ref. Grade 5) | 0.24 | 0.08–0.75 | 0.014 | 1.35 | 0.52–3.48 | 0.540 |
| Visit (ref. Follow up 1) | 1.01 | 0.21–4.92 | 0.989 | 0.74 | 0.31–1.76 | 0.494 |
| Distance to water body class (ref. < 1 km) | | | | | | |
| 1–2 km | 0.11 | 0.03–0.34 | <0.001 | | | |
| 2–3 km | 0.18 | 0.04–0.85 | 0.031 | | | |
| >3 km | 0.08 | 0.01–0.54 | 0.010 | | | |
| Household wealth index (ref. 1st quintile) | | | | | | |
| 2 | 3.75 | 0.49–28.7 | 0.203 | | | |
| 3 | 0.38 | 0.08–1.83 | 0.233 | | | |
| 4 | 3.22 | 0.63–16.3 | 0.159 | | | |
| 5 | 1.71 | 0.03–1.70 | 0.432 | | | |
| Square root of slope | 0.77 | 0.45–1.32 | 0.340 | | | |
| Baseline intensity of infection (ref. Light infection) | 2.59 | 0.81–8.30 | 0.110 | | | |
| *Alpha* (overdispersion parameter) | | | | *22.6* | *17.9–28.3* | *<0.001* |

on *Schistosoma haematobium* preventive chemotherapy (*Saathoff et al., 2004*; *WHO Expert Committee on the Control of Schistosomiasis, 2002*) under strict supervision of trained nurses. Children who had false negative urine reagent strip results (based on microscopy gold standard) were subsequently traced back in school, and became eligible for treatment and inclusion in the cohort analysis. However, because of the time lag between laboratory testing and treatment, some children with a false negative test results missed treatment for baseline infection if they were absent from school during tracing and were therefore not eligible for inclusion in the cohort analysis. In addition, some parents/guardians of children who were screened and treated for baseline infection did not give consent for their children to participate in the follow-up rounds (*Figure 1*).

Therefore, the re-infection cohort was defined as children who had microscopically confirmed *Schistosoma haematobium* infection at baseline, were treated for baseline infection and consented to participate in at least one follow-up round. Eligibility for the second round of follow-up was subject to treatment at round 1 (for those found to be infected during screening) and consent

at round 2 regardless of the infection status at round 1, or children who were treated for baseline infection and consented for only round 2 of screening and treatment.

## Sample collection and laboratory analysis

Sample collection, consenting and laboratory testing procedures were similar to those conducted during the baseline parasitological survey that we reported previously (*Tanser et al., 2018*). Urine samples were collected between 10:00 and 12:00 hr using 500 mL honey jars (one sample per child). Each sample testing for infection was done in two stages: 1) testing using urinalysis reagent strips (Bayer Uristix) in schools and 2) confirmatory microscopy analysis in the laboratory. The urine samples were aliquoted and processed in duplicates of 10 mL sub-samples and diluted using 2% methiolate in 5% Formalin. Filtration was done using the polycarbonate filters (diameter of 25 mm and 8.0 µm pore size). Upon sample filtration, each urine filter, potentially containing *Schistosoma haematobium* eggs, was placed on a glass slide, stained and examined under a microscope with x10 magnification. Eggs were counted by trained laboratory technicians and expressed per 10 mL of urine. The primary outcome of interest was the subject level egg count determined by averaging the duplicate egg counts for each participating child in each follow-up round. Here, infection or re-infection status refers to microscopically confirmed *Schistosoma haematobium* eggs in urine samples. We categorized egg counts into a priori groups of heavy ($\geq$50 eggs/10 mL) and light (<50 eggs/10 mL) re-infections following WHO classification guidelines (*WHO Expert Committee on the Control of Schistosomiasis, 2002*) on intensity of re-infection with *Schistosoma haematobium*.

## Data analysis

Descriptive statistics were used to describe demographic and environmental characteristics of *Schistosoma haematobium* re-infection. We used geometric means to summarize egg counts and estimated disease burden by computing prevalence with 95% confidence intervals (95% CI). Royston's $\chi^2$-test-of-trend and the classical chi-square test for nominal data were used to assess the linear trend and nominal associations between variables respectively.

## Re-infection rates

Given that the exact date of re-infection after treatment is not observed, we randomly imputed a re-infection date between the treatment date and the testing date assuming a uniform distribution (*Vandormael et al., 2018*) for all children who were re-infected with *Schistosoma haematobium* in both rounds of follow-up. We used the imputed dates of re-infection or testing date for non-infected children to compute time at risk that was used as a denominator when computing the 2007–2008 re-infection rates and computed the 95% CIs assuming the Poisson distribution.

## Covariates for schistosomiasis re-infection

We have previously presented a detailed description of the procedure used to derive the exposure variable and potential confounders (*Tanser et al., 2018*). Water supply from the reservoir to the study area is mainly through gravitated PVC pipes. A household had access to safe water supplies if there was reliable piped water in the dwelling or if the key household informant reported that the household used water from the public tab, borehole, protected dug well, protected spring or rainwater from storage tanks, for the household chores. We derived the community piped water coverage for each residential homestead of the participating child from a weighted two-dimensional Gaussian kernel of 2 km radius using data from the 2007 homestead level survey on access to piped water (*Figure 6B*). The 2 km radius was selected based on our previous analysis in which a tradeoff between sensitivity to local variations and robustness to random noise was considered (*Tanser et al., 2018*). We used the derived community piped water coverage (exposure variable), and the well-established potential confounders (environmental and social economic covariables [*Tanser et al., 2018*; *Lai et al., 2015*; *Brooker, 2007*; *Appleton, 1978*; *Simoonga et al., 2009*; *Clennon et al., 2004*]) in both descriptive and inferential analysis. We also obtained the altitude, slope, distance to the nearest water body, landcover classification and household wealth index as we previously described (*Tanser et al., 2018*).

## Regression modelling

Count models have previously been utilized to assess determinants of host intensity of re-infection and account for host heterogeneity inherent in the distribution of schistosomiasis egg counts (*Chipeta et al., 2014*). The negative binomial model was the best fit to our data among the plausible count models that we examined in a data driven model selection procedure (*Tang et al., 2012*). We used univariable analyses (retaining significant covariates at p<0.1) and backwards exclusion of non-significant covariates (p>0.05) to arrive at a final parsimonious model. The exposure variable (community piped water coverage), age, toilet and sex were included in multivariable models regardless of their significance in the univariable analyses. We used predictive margins to estimate egg counts at various predefined levels of the exposure variable from the final multivariable regression model. To account for correlation among children who contributed data to both the 1st and 2nd follow-up rounds (pooled analysis), we obtained robust standard errors in models adjusted for the fixed effects of follow-up round and further conducted subgroup analyses on each follow-up round separately.

We performed statistical analyses using Stata 14 (Stata Corp, College Station, TX, USA) and computed the spatial scan statistics (*Kulldorff et al., 2009*) using SaTScan version 9.4.2 software (Havard medical School, Boston, MA, USA). We used ArcGIS version 10.3 (ESRI, Redlands, CA, USA) (*Esri and Here, 2004*) for the standard Gaussian kernel density estimation and cartographic display.

## Local cluster detection

In our baseline analysis (*Tanser et al., 2018*), we described geographical heterogeneity of *Schistosoma haematobium* infection prevalence and identified 4 clusters of significantly high relative risk. Here, we used the Gaussian kernel density estimations to describe the geographical heterogeneity of re-infection intensity and identified the presence of significant geographical clusters of intense re-infection using the scan statistic implemented in SaTScan software. Briefly, we examined for geographical areas experiencing significantly higher intensity of re-infection (mean log-egg count) than would be expected by chance using the normal probability model (*Kulldorff et al., 2009*). SaTScan software imposes a scanning window (predefined here to be circular and non-overlapping) that moves systematically across geographical space and with varying radius. For each geographical location and scanning window size, the geometric mean egg count is computed, and significance testing performed using the Monte-Carlo simulation.

## Role of the funding source

The funders had no role in the study design, data collection, analysis, interpretation of results, manuscript writing or decision to submit for publication. The corresponding author had full access to the data and made the final decision to submit for publication after obtaining approval from the coauthors.

# Acknowledgements

We thank the primary school children for their willingness to participate in this study. We thank the children's parents and/or guardians, school principals and teachers for the assistance in conducting the study. The authors are indebted to the study staff at AHRI and School-Health team at Hlabisa Hospital for their invaluable assistance in conducting the parasitological survey. The authors wish to express their grateful thanks to Colleen Archer (University of KwaZulu-Natal) for conducting the microscopic analysis and Colin Newell for database support. The study was funded through the National Institute of Health via the International Collaboration in Infectious Disease Research (ICIDR). The Africa Health Research Institute is funded by the Wellcome Trust, UK.

# Additional information

### Funding

| Funder | Grant reference number | Author |
| --- | --- | --- |
| National Institutes of Health | International Collaboration in Infectious Disease Research (ICIDR) | Christopher Appleton Frank Tanser |

| Wellcome Trust | Africa Health Research Institute | Frank Tanser |
|---|---|---|

The funders had no role in study design, data collection and interpretation, or the decision to submit the work for publication.

## Author contributions

Polycarp Mogeni, Conceptualization, Data curation, Formal analysis, Validation, Visualization, Writing - original draft, Writing - review and editing; Alain Vandormael, Conceptualization, Formal analysis, Writing - original draft, Writing - review and editing; Diego Cuadros, Formal analysis, Visualization, Writing - original draft, Writing - review and editing; Christopher Appleton, Conceptualization, Funding acquisition, Investigation, Methodology, Writing - original draft, Project administration, Writing - review and editing; Frank Tanser, Conceptualization, Resources, Data curation, Formal analysis, Supervision, Funding acquisition, Validation, Investigation, Methodology, Writing - original draft, Project administration, Writing - review and editing

## Author ORCIDs

Polycarp Mogeni (iD) https://orcid.org/0000-0003-1926-7576
Alain Vandormael (iD) http://orcid.org/0000-0002-5742-0511
Frank Tanser (iD) http://orcid.org/0000-0001-9797-0000

## Ethics

Human subjects: Ethical approval was provided by the Biomedical Research Ethics Committee of the university of KwaZulu-Natal (reference #E165/05). Written informed consent was sought from parents or guardians of the participating children for both rounds of follow-up in 2007 and 2008 and assent obtained from the children during the follow-up surveys.

## Decision letter and Author response

Decision letter https://doi.org/10.7554/eLife.54012.sa1
Author response https://doi.org/10.7554/eLife.54012.sa2

## Additional files

### Supplementary files

• Source data 1. Prevalence of re-infection, intensity of re-infection and re-infection rate (per 100-person year of follow-up) among individuals treated at baseline for *S. haematobium* infection.

• Supplementary file 1. Predictors of *Schistosoma haematobium* re-infection using data from follow up round 1 only. Model 1 presents results from a univariable negative binomial model and Model 2 presents results from a multivariable negative binomial model (N = 253).

• Supplementary file 2. Predictors of *Schistosoma haematobium* re-infection using data from follow up round 2 only. Model 1 presents results from a univariable negative binomial and Model 2 presents results from a multivariable negative binomial model (N = 125).

• Supplementary file 3. Characteristics of participants who dropped out of the study at follow-up round 1. Piped water coverage (exposure variable) was similar between participants who dropped out of the study and those that were enrolled and examined. Significantly higher dropouts were only observed among participants residing further from water bodies. Piped water coverage was derived from the Gaussian kernel density estimation of radius three kilometers.

• Supplementary file 4. Characteristics of participants who dropped out of the study at follow-up round 2. Piped water coverage (exposure variable) was similar between participants who dropped out of the study and those that were enrolled and examined. Significantly higher dropouts were only observed among girls. Piped water coverage was derived from the Gaussian kernel density estimation of radius three kilometers.

• Transparent reporting form

## Data availability

The datasets used for the analysis presented in this study are available from the Africa Health Research Institute (AHRI) data repository https://data.africacentre.ac.za/index.php/auth/login/?destination=. To access the licensed datasets, the applicant must agree to the terms and conditions of use by completing an Application for Access to a Licensed Dataset. This request will be reviewed by the AHRI Data Release Committee, who may decide to approve the request, to deny access to the data, or to request additional information from the applicant.

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
