## [Decision Letter]

Thank you for submitting your article "Impact of community piped water coverage on re-infection with urogenital schistosomiasis in rural South Africa" for consideration by *eLife*. Your submission is a Research Advance; this format is for substantial developments that directly build upon a previous *eLife* paper. Specifically, the present submission is intended to expand on a paper by your colleagues entitled "Impact of the scale-up of piped water on urogenital schistosomiasis infection in rural South Africa", authored by Tanser et al.

With the above context in mind, your article has been reviewed by two peer reviewers, and the evaluation has been overseen by a Senior/Reviewing Editor. The reviewers have opted to remain anonymous.

As is customary in *eLife*, the reviewers have discussed their reviews with one another. What follows below is an edited compilation of the essential and ancillary points provided by reviewers in their critiques and in their interaction post-review. Although we expect that you will address these comments in your response letter we also need to see the corresponding revision in the text of the manuscript. Some of the reviewers' comments may seem to be simple queries or challenges that do not prompt revisions to the text. Please keep in mind, however, that readers may have the same perspective as the reviewers. Therefore, it is essential that you attempt to amend or expand the text to clarify the narrative accordingly.

Summary:

The manuscript entitled "Impact of community piped water coverage on re-infection with urogenital schistosomiasis in rural South Africa" presents a study with new results supporting the scale-up of piped water coverage against *Schistosoma haematobium* transmission. Nested in a broader project that was published last year in *eLife*, this longitudinal study shows that a one-percent increase in piped water coverage was associated with 4.4% decline in re-infection intensity. These are relevant public health results that could improve the control of an important though neglected tropical disease.

Essential revisions:

1) Although the authors referred whenever possible to their previous article, it should also be possible to read the current manuscript as a self-contained work. Please provide one or two sentences of background on *Schistosoma haematobium* as a clinical and epidemiological entity.

2) You demonstrate a 15.4% reinfection after one round of praziquantel mass drug administration, and 12.8% reinfection rate after the second round of MDA. While your main findings are focused on re-infection intensity, this is but one metric to focus on. Relatedly, re-infection rates are also extremely important. What is the significance of piped water contributing to a 4.4% decline in infection intensity? Infection intensity may be driven by many factors, including degree of exposure, questionable development of protective immunity, and time since the most recent praziquantel treatment. There is ample evidence (e.g. Taabo District, Cote d'Ivoire, among many other sites), where MDA will decrease environmental contamination of water sources and partially block the lifecycle of the parasite. Over only a few years, we expect to see a decline in infection rates plus after even 2 rounds of MDA (typically annually or biannually) most infections are <50 eggs per 10 cc urine (low intensity as per WHO guidelines). Please clarify why the infection rates are similar after two rounds of MDA, and why there is only a very modest drop in infection intensity given the piped water intervention plus MDA.

3) It is unclear how many observations were included in the results of the analysis presented in the different figures and tables. It is difficult to assess the internal validity of the results and how they were affected by attrition.

4) Introduction, last paragraph: the number of children included in the follow-up round should be presented in the Results section. Indeed only few details can be given in this section. As a result readers could be misled to think that 333 children took part in both rounds.

Results:

5) Information is lacking in certain figures; please expand the figure legends.

6) "In the pooled adjusted negative binomial model…" should the readers assume that 378 measurements were included in this analysis?

7) Subsection “Impact of community piped water coverage on intensity of re-infection”: Figure 4 and Figure 2B do not clearly give a measure of the association between mean egg counts and piped water coverage, the authors should consider to mention these results explicitly in the Results section.

8) Figure 5: Consider replacing the study area in a map of South-Africa.

Discussion:

9) Discuss attrition and how power may have been impaired by the size of your study sample. Could differential dropout have biased your results?

10) "… or whether the drug given was taken in all situations". Are you suggesting that the praziquantel treatment might not have been taken by all children? Was the treatment administration not supervised?

11) Discussion, last paragraph: a cohort is by essence longitudinal.

12) Discussion, last paragraph: please italicize [*Schistosoma haematobium*]

Materials and methods

13) Subsection “Re-infection cohort design”: it is unclear how children could have a false negative RDT for the baseline infection and yet been included. Please clarify.

14) Subsection “Sample collection and laboratory analysis”: The authors discuss using RDT testing with test strips. I presume they are referring to "urine reagent strips". RDT may be misinterpreted for the RDT point-of-care test for *S. mansoni* (the POC-CAA test). Please rephrase this to state that these are urine reagent strips, and that you are evaluating for hemoglobinuria.

15) Subsection “Sample collection and laboratory analysis”: More detail is required about urine processing. What size filters were used? How many samples were provided by each child? What time of day was urine collected? Traditionally this is between 10:00 and 14:00 to maximize egg collection due to circadian variability in excretion.

---

## [Author Response]

Essential revisions:1) Although the authors referred whenever possible to their previous article, it should also be possible to read the current manuscript as a self-contained work. Please provide one or two sentences of background on *Schistosoma haematobium* as a clinical and epidemiological entity.

We have included a paragraph in the Introduction giving a summary of the epidemiology as shown below.

“About 243 million people are infected with schistosomiasis worldwide, of whom ~93% reside in sub-Saharan Africa where children carry the greatest burden of the disease. […] However, *Schistosoma haematobium*has the widest geographical coverage in SSA and is the main cause of infection inthe Hlabisa sub-district, where our study is based[1].”

2) You demonstrate a 15.4% reinfection after one round of praziquantel mass drug administration, and 12.8% reinfection rate after the second round of MDA. While your main findings are focused on re-infection intensity, this is but one metric to focus on. Relatedly, re-infection rates are also extremely important. What is the significance of piped water contributing to a 4.4% decline in infection intensity? Infection intensity may be driven by many factors, including degree of exposure, questionable development of protective immunity, and time since the most recent praziquantel treatment.

We agree that re-infection rates are another important outcome metric. However, re-infection intensity provides a more precise and nuanced outcome measure, which is therefore often used to quantify the effect of interventions in schistosomiasis research [2-7] and to map transmission intensity [8, 9].

Our cohort study, by design, only included individuals who were at a high risk of re-infection (i.e. only participants who had confirmed *Schistosoma haematobium* infection at baseline) and the use of a less precise binary measure like re-infection, whilst potentially useful would require a larger sample size to be able to detect significant effects. We do recognize that re-infection intensity is a function of a set of features as pointed out by the reviewers. In this respect we adjusted for exposure time since recent treatment and age (a proxy for acquisition of immunity) in the regression models. Time since recent treatment, was not a significant predictor in the model partly because the follow-up timepoints were predefined. Although immunity is important, it may not play a major role in this study given the limited age range observed. Therefore, we predict that the intensity of infection observed was due to the varying degree of exposure and that high piped water coverage significantly decreases exposure resulting to lower intensity of infection.

There is ample evidence (e.g. Taabo District, Cote d'Ivoire, among many other sites), where MDA will decrease environmental contamination of water sources and partially block the lifecycle of the parasite. Over only a few years, we expect to see a decline in infection rates plus after even 2 rounds of MDA (typically annually or biannually) most infections are <50 eggs per 10 cc urine (low intensity as per WHO guidelines). Please clarify why the infection rates are similar after two rounds of MDA, and why there is only a very modest drop in infection intensity given the piped water intervention plus MDA.

The reviewer raises an important point here. Unlike most MDA studies (which target all children regardless of their infection status and include all children treated at baseline in the denominator when computing re-infection rates), our re-infection cohort participants were children who were infected and were treated for baseline infection (the denominator only includes children who were infected, treated at baseline and examined at any given follow-up round). Therefore, our cohort participants were children who were at a higher risk of re-infection given their baseline status. The prevalence of re-infection at round 1 and 2 would have been much lower if the entire cohort were to be examined and included in the denominator (as done in MDA) given their predicted low exposure to infection. We also note that, our study only tested and treated grade 5 and 6 children. Thus, there were still many untreated and infected children of other ages capable of sustaining the transmission cycle in these communities. This ultimately results in a high potential for re-infection in our cohort of previously infected children.

We include the following statement in the Discussion section:

“Finally, we recommend further studies to examine the impact of an integrated approach [10] that include MDA, piped water coverage and behavioral change education (potentially including installation of safe water recreational areas [11]) on re-infection.”

3) It is unclear how many observations were included in the results of the analysis presented in the different figures and tables. It is difficult to assess the internal validity of the results and how they were affected by attrition.

We have included the number of observations in both the figures and table legends where appropriate and included them in tables as well. For instance:

**“**Figure 2: Histograms of *Schistosoma haematobium* egg counts and community piped water coverage for the re-infection cohort participants (n=378). […] Community piped water coverage in the community surrounding each child was derived from the population-based 2007 piped water use survey conducted among all households in the study area.”

“Figure 4: Impact of piped water coverage in the community surrounding each child’s residence on *Schistosomahaematobium* re-infection intensity. The margin plot was constructed from the final parsimonious multivariable negative binomial regression model for the pooled dataset (n=378, incidence rate ratio = 0.96, P=0.002). Piped water coverage was estimated using the Gaussian kernel density methodology.”

“Table 2: Predictors of intensity of re-infection with *Schistosomahaematobium* (pooled analysis, n=378). […] Homestead level piped water coverage was derived from a Gaussian kernel density estimation using data from a survey conducted in 2007.”

4) Introduction, last paragraph: the number of children included in the follow-up round should be presented in the Results section. Indeed only few details can be given in this section. As a result readers could be misled to think that 333 children took part in both rounds.

We agree with the reviewers’ comments and delete the number of children from this section. We provide a detailed breakdown of the number of children participating in each round on the flowchart presented as Figure 1. See Figure 1 legend.

Results:5) Information is lacking in certain figures; please expand the figure legends.

We have expanded the figure legends to clearly show the sample size and other missing information.

“Figure 2: Histograms of *Schistosomahaematobium* egg counts and community piped water coverage for the re-infection cohort participants (n=378). […] Community piped water was derived from 2007 piped water use survey conducted in the study area.”

“Figure 4: Impact of piped water coverage in the community surrounding the child’s residence on *Schistosoma haematobium* re-infection intensity. The margin plot was constructed from the final parsimonious multivariable negative binomial regression model for the pooled dataset (n=378, incidence rate ratio = 0.96, P=0.002). Piped water coverage was estimated using the Gaussian kernel density methodology.”

6) "In the pooled adjusted negative binomial model…" should the readers assume that 378 measurements were included in this analysis?

We have added the sample size as shown below: see the first paragraph of the subsection “Impact of community piped water coverage on intensity of re-infection”.

7) Subsection “Impact of community piped water coverage on intensity of re-infection”: Figure 4 and Figure 2B do not clearly give a measure of the association between mean egg counts and piped water coverage, the authors should consider to mention these results explicitly in the Results section.

We have expanded the legend for Figure 4 to clearly show the association as derived from the final negative binomial model (n=378). The graph shows the predicted margins from the final multivariable model. We include the information in the figure legend.

**“**Figure 4: Impact of piped water coverage in the community surrounding the child’s residence on *Schistosoma haematobium* re-infection intensity. […] Piped water coverage was estimated using the Gaussian kernel density methodology.”

We have also deleted reference to Figure 2B from the following statement:

“Lower piped water coverage areas were associated with significantly higher mean egg counts albeit with relatively low numbers of children living at low piped water coverages (Figure 4)”.

8) Figure 5: Consider replacing the study area in a map of South-Africa.

In the Materials and methods section, we have included the map of South Africa (Figure 6) showing the study area. We have also included the study area showing the major roads and piped water coverage for the year 2007.

Discussion:9) Discuss attrition and how power may have been impaired by the size of your study sample. Could differential dropout have biased your results?

Attrition will inevitably lead to reduced power to detect meaningful associations. We however demonstrated a significant association between piped water coverage and re-infection intensity. We have included Supplementary files 3 and 4, and a paragraph discussing the potential bias attributable to attrition in our analysis and argued that the potential bias is likely towards the null and thus our effect sizes are likely marginally conservative.

We include the following paragraph in the Results section:

“Details of cohort retention are provided in Supplementary file 3 and Supplementary file 4. […] However, there was no clear pattern between dropout and piped water coverage.”

We also discuss the limitation in the Discussion section:

“Whilst we demonstrated a clear relationship between piped water coverage and re-infection intensity, we were not powered to detect differences in the rate of re-infection by piped water coverage category. […] Therefore, a higher proportion of attrition among girls will potentially bias the association of piped water on re-infection intensity towards the null hypothesis and implying our effect estimates are conservative.”

10) "… or whether the drug given was taken in all situations". Are you suggesting that the praziquantel treatment might not have been taken by all children? Was the treatment administration not supervised?

We agree that the statement is unclear. We confirm that treatment administration was supervised and delete the statement from the Discussion section.

We also spell this out in the Materials and methods section:

“A single oral dose of 40 mg/kg body weight of praziquantel was administered to children with detectable microhematuria following recommended World Health Organization (WHO) guidelines on *Schistosoma haematobium* preventive chemotherapy [1, 12] under strict supervision of trained nurses.”

11) Discussion, last paragraph: a cohort is by essence longitudinal.

We have deleted the word “longitudinal”

12) Discussion, last paragraph: please italicize [*Schistosomahaematobium*]

We have italicized “*Schistosoma haematobium*”

Materials and methods13) Subsection “Re-infection cohort design”: it is unclear how children could have a false negative RDT for the baseline infection and yet been included. Please clarify.

Thank you for this question. Urine reagent strips (changed from RDT given the highlighted potential confusion) were used in schools to determine children who were infected and therefore eligible for treatment immediately in school. In addition, we conducted a confirmatory microscopy analysis in our laboratories. Children who had false negative results by urine reagent strips at baseline were traced back in school and treated and were therefore eligible for inclusion in the cohort analysis. However, false negative children that were not treated at baseline (absent in school during tracing) were not eligible for inclusion in the cohort analysis.

We have included the following statements:

“Children who had false negative urine reagent strip results (based on microscopy gold standard) were subsequently traced back in school, treated and included in the cohort analysis. However, because of the time lag between laboratory testing and treatment, some children with a false negative test results missed treatment for baseline infection if they were absent from school during tracing and were therefore not eligible for inclusion in the cohort analysis.”

14) Subsection “Sample collection and laboratory analysis”: The authors discuss using RDT testing with test strips. I presume they are referring to "urine reagent strips". RDT may be misinterpreted for the RDT point-of-care test for S. mansoni (the POC-CAA test). Please rephrase this to state that these are urine reagent strips, and that you are evaluating for hemoglobinuria.

We have changed this, see the Materials and methods section:

“*Schistosomahaematobium* infection status was determined using the urine reagent strips for evaluating microhematuria with confirmation using microscopy diagnosis.”

15) Subsection “Sample collection and laboratory analysis”: More detail is required about urine processing. What size filters were used? How many samples were provided by each child? What time of day was urine collected? Traditionally this is between 10:00 and 14:00 to maximize egg collection due to circadian variability in excretion.

We have included this information in the Materials and methods section:

“Urine samples were collected between 10:00 and 12:00 hours using 50 mL conical tubes (one sample per child). Each sample testing for infection was done in two stages: 1) testing using urinalysis reagent strips (Bayer Uristix) in schools and 2) confirmatory microscopy analysis in the laboratory. […] Eggs were counted by trained laboratory technicians and expressed per 10 mL of urine.

References

1) Saathoff, E., et al., Patterns of *Schistosoma haematobium* infection, impact of praziquantel treatment and re-infection after treatment in a cohort of schoolchildren from rural KwaZulu-Natal/South Africa. BMC Infectious Diseases, 2004. 4(1): p. 40.

2) Anderson, R.M., et al., What is required in terms of mass drug administration to interrupt the transmission of schistosome parasites in regions of endemic infection? Parasites and vectors, 2015. 8: p. 553-553.

3) Bah, Y.M., et al., Schistosomiasis in School Age Children in Sierra Leone After 6 Years of Mass Drug Administration With Praziquantel. Frontiers in public health, 2019. 7: p. 1-1.

4) Chandiwana, S.K., M.E. Woolhouse, and M. Bradley, Factors affecting the intensity of reinfection with *Schistosoma haematobium* following treatment with praziquantel. Parasitology, 1991. 102 Pt 1: p. 73-83.

5) Knopp, S., et al., A 5-Year intervention study on elimination of urogenital schistosomiasis in Zanzibar: Parasitological results of annual cross-sectional surveys. PLoS neglected tropical diseases, 2019. 13(5): p. e0007268-e0007268.

6) Phillips, A.E., et al., Assessing the benefits of five years of different approaches to treatment of urogenital schistosomiasis: A SCORE project in Northern Mozambique. PLoS neglected tropical diseases, 2017. 11(12): p. e0006061-e0006061.

7) Toor, J., et al., The design of schistosomiasis monitoring and evaluation programmes: The importance of collecting adult data to inform treatment strategies for *Schistosoma mansoni*. PLoS neglected tropical diseases, 2018. 12(10): p. e0006717-e0006717.

8) Vounatsou, P., et al., Bayesian geostatistical modelling for mapping schistosomiasis transmission. Parasitology, 2009. 136(13): p. 1695-1705.

9) Chadeka, E.A., et al., A high-intensity cluster of *Schistosoma mansoni* infection around Mbita causeway, western Kenya: a confirmatory cross-sectional survey. Tropical medicine and health, 2019. 47: p. 26-26.

10) Knopp, S., et al., Evaluation of integrated interventions layered on mass drug administration for urogenital schistosomiasis elimination: a cluster-randomised trial. The Lancet. Global health, 2019. 7(8): p. e1118-e1129.

11) Kosinski, K.C., et al., Effective control of *Schistosoma haematobium* infection in a Ghanaian community following installation of a water recreation area. PLoS neglected tropical diseases, 2012. 6(7): p. e1709-e1709.

12) WHO, E., Committee, Prevention and control of schistosomiasis and soil-transmitted helminthiasis. World Health Organ Tech Rep Ser, 2002. 912: p. i-vi, 1-57, back cover.